# miR-143-3p Promotes Ovarian Granulosa Cell Senescence and Inhibits Estradiol Synthesis by Targeting UBE2E3 and LHCGR

**DOI:** 10.3390/ijms241612560

**Published:** 2023-08-08

**Authors:** Jingxian Deng, Yan Tang, Lu Li, Rufei Huang, Zhaoyang Wang, Tao Ye, Ziyan Xiao, Meirong Hu, Siying Wei, Yuxin Wang, Yan Yang, Yadong Huang

**Affiliations:** 1Department of Pharmacology, Jinan University, Guangzhou 510632, China; dengjingxian@stu2021.jnu.edu.cn; 2Department of Cell Biology, Jinan University, Guangzhou 510632, Chinalilu2022@stu2022.jnu.edu.cn (L.L.); sophie12@stu2022.jnu.edu.cn (R.H.); wzy1003@stu2021.jnu.edu.cn (Z.W.); taoyhust@stu2022.jnu.edu.cn (T.Y.); xzy2021@stu2021.jnu.edu.cn (Z.X.); 358070hmr@stu2021.jnu.edu.cn (M.H.); wsy1844@stu2021.jnu.edu.cn (S.W.); yxwang@stu2021.jnu.edu.cn (Y.W.); 3Guangdong Province Key Laboratory of Bioengineering Medicine, Guangzhou 510632, China

**Keywords:** miR-143-3p, senescence, estradiol, UBE2E3, LHCGR

## Abstract

The ovary is a highly susceptible organ to senescence, and granulosa cells (GCs) have a crucial role in oocyte development promotion and overall ovarian function maintenance. As age advances, GCs apoptosis and dysfunction escalate, leading to ovarian aging. However, the molecular mechanisms underpinning ovarian aging remain poorly understood. In this study, we observed a correlation between the age-related decline of fertility and elevated expression levels of miR-143-3p in female mice. Moreover, miR-143-3p was highly expressed in senescent ovarian GCs. The overexpression of miR-143-3p in GCs not only hindered their proliferation and induced senescence-associated secretory phenotype (SASP) but also impeded steroid hormone synthesis by targeting ubiquitin-conjugating enzyme E2 E3 (*Ube2e3*) and luteinizing hormone and human chorionic gonadotropin receptor (*Lhcgr*). These findings suggest that miR-143-3p plays a substantial role in senescence and steroid hormone synthesis in GCs, indicating its potential as a therapeutic target for interventions in the ovarian aging process.

## 1. Introduction

The swift pace of economic development and the rising status of women in the workforce have contributed to a rise in the average age of childbirth across the globe [1]. The success of pregnancy is significantly impacted by a woman’s age, irrespective of whether the conception occurs naturally or with the aid of assisted reproductive technology [2,3]. Studies have indicated that the aging process of women affects the ovaries initially [4,5]. At around age 30, ovarian function starts to decline, leading to reduced hormonal secretion and ovulation disorders, which can make it challenging for women to conceive [6]. As a result, prolonging the lifespan of women’s ovaries is a pressing issue that requires attention. However, the molecular mechanisms responsible for ovarian aging are not yet fully understood.

The follicle is considered the fundamental functional unit of the ovary, comprised of oocytes, GCs, and follicular membrane cells. GCs play a crucial role in facilitating oocyte development and supporting overall ovarian function [7]. As vital somatic cells within the ovarian follicle, GCs envelop and encase the oocyte, providing a protective barrier and facilitating cellular communication between the oocyte and the extracellular environment [8,9]. As GCs proliferate, they form multiple layers around the oocyte, leading to its maturation and eventual ovulation [10]. Additionally, GCs play a significant role in synthesizing and secreting steroid hormones, which are crucial regulators of reproductive function and under tight control by the hypothalamic–pituitary–gonadal (HPG) axis [11,12]. Receptors in the GCs membrane, including follicle-stimulating hormone receptor (FSHR) and luteinizing hormone/choriogonadotropin receptor (LHCGR), bind to the corresponding pituitary-secreted gonadotropins [13]. Both FSHR and LHCGR play critical roles in facilitating successful follicular development and ovulation. FSHR is a G protein-coupled receptor that binds to FSH and is primarily expressed in the GCs of the ovary. The activation of FSHR promotes the growth and development of ovarian follicles and stimulates estradiol (E2) production [14]. LHCGR is also a G protein-coupled receptor that binds to luteinizing hormone (LH) and human chorionic gonadotropin (hCG). LH and hCG can both bind to LHCGR, leading to the activation of downstream signaling pathways that stimulate the production of progesterone (P4) and the process of ovulation [15]. Dysregulation of these receptors can lead to infertility and other reproductive disorders. Research showed that the decrease in *Lhcgr* expression in GCs causes abnormal follicle development and leads to ovulation failure [16]. 

The quality and quantity of follicles determine the function and lifespan of the ovary [17]. GCs’ proliferation and function are crucial in follicular development, maturation, and atresia. During follicular development, GCs proliferate rapidly and play a crucial role in supporting and sustaining oocyte growth through their nutritional support and paracrine functions [18]. Thus, excessive apoptotic activity in GCs can alter the follicular microenvironment, leading to abnormal follicular development and reduced oocyte quality [19]. As age increases, the number of senescent GCs in the ovary also increases. Aging GCs exhibit dysregulated signaling pathways involved in various cellular processes. For instance, the accumulation of a significant amount of reactive oxygen species (ROS) in GCs inhibited Wnt16 signaling, leading to increased β-catenin activity and subsequently promoting GC aging [20]. Decreased activity of the PI3K/Akt pathway inhibited GC proliferation and accelerated GC apoptosis [21]. Studies have shown that as women ages, the structure and function of mitochondria within GCs undergo changes, including a reduction in the number of mitochondria, accumulation of mitochondrial DNA damage, and a decline in mitochondrial respiratory chain activity. These changes may result in compromised follicular development and diminished oocyte quality [22]. Age-related changes in the endocrine and paracrine factors that regulate folliculogenesis are also associated with increased rates of apoptosis in GCs [23]. These changes ultimately lead to a decline in ovarian function and reduced fertility. Therefore, understanding the mechanisms involved in age-related GC apoptosis is crucial for developing interventions to improve reproductive health and fertility in aging women. It has been reported that chronic inflammation is found in aging ovaries, and this chronic inflammation is manifested by an increase in multiple pro-inflammatory transcription factors, cytokines, and chemokines [24]. Age-dependent increases in ovaries inflammation may directly or indirectly affect gamete quantity and quality [25]. For instance, elevated levels of TNF-α in follicular fluid are associated with poor oocyte quality. Ovaries of mice knocked out of IL-1β and IL-1α exhibit more vigorous fertility [26]. Although it is known that the incidence of GC apoptosis increases with age, the mechanisms underlying these processes are not yet fully understood. Therefore, further studies are needed to identify the key factors and signaling pathways involved in GC apoptosis with aging.

MicroRNAs (miRNAs) are short (about 21 nucleotides; nt) RNA sequences encoded by endogenous genes. miRNAs regulate gene expression by binding to the mRNA 3′ untranslated region (3′ UTR) [27]. miRNA-expression profiles vary between disease states and different tissues, allowing miRNAs to be biomarkers of disease [28]. In GCs, miRNA-205 promoted apoptosis and inhibited estradiol synthesis by targeting CREB1 [29]. miRNA-484-regulating GCs function via Yes-associated protein isoform 1 (YAP1)-mediated mitochondrial dysfunction and apoptosis [30]. The intronic microRNA let-7g targets and downregulates transforming growth factor-β type 1 receptor (TGFBR1) to inhibit the TGF-β/SMAD signaling pathway, thereby promoting GC apoptosis [31].

In this study, we observed a significant increase in the expression of miR-143-3p in the ovaries and serum of aged mice. miR-143-3p has been reported to be involved in regulating tumor development and cardiovascular diseases [32,33]. In GCs, miR-143-3p targeted and downregulated FSHR expression during follicular atresia [14]. However, the role of miR-143-3p in GC senescence remains poorly understood. Our study revealed that overexpression of miR-143-3p promotes cellular senescence and inhibits steroid hormone synthesis in GCs. These findings provide new insights into the regulatory pathways that control GCs senescence and offer a promising approach to understanding and treating age-related reproductive disorders.

## 2. Results

### 2.1. miR-143-3p Is Highly Expressed in the Ovaries of Aged Mice

We observed an increase in the expression of miR-143-3p in ovaries of mice at 15 months of age (old group) (Figure 1A). The level of miR-143-3p in the ovaries of old mice was significantly increased compared to both the young (4 months) and middle-aged (8 months) mice (Figure 1B). Furthermore, a significant increase in the expression level of miR-143-3p in the serum was observed, with the same pattern in the ovaries (Figure 1C). 

As levels of miR-143-3p increase, changes occur in the reproductive ability of aged mice, indicated by a significant decrease in litter numbers of females at 15 months of age (Figure 1D). As a result, we investigated the serum steroid hormone levels of mice of different age groups and revealed that E2 and P4 levels significantly decreased in older mice compared to young and middle-aged mice (Figure 1E,F). Fibrosis of the ovaries and the number of β-galactosidase staining-positive cells in the ovary significantly increased with age (Figure 1G,H).

These finding suggest that female mice experience a decline in fertility and steroid hormones levels as they age, which is accompanied by an increased expression of miR-143-3p in their ovaries. This raises the question of whether miR-143-3p play a role in the age-related decline in female reproductive capacity.

### 2.2. miR-143-3p Inhibits E2 and P4 Synthesis in GCs

We isolated and subcultured primary GCs from the ovaries of mice to explore the involvement of miR-143-3p in aging-associated subfertility in females and the decline in E2 and P4 levels. The purity of the GCs was determined using FSHR immunofluorescence, which showed that FSHR was expressed in over 95% of the isolated cells (Figure 2A). β-galactosidase staining was performed in the early passage and late-passage primary GCs to confirm their senescence. Our findings revealed that the number of β-galactosidase staining-positive cells in GCs passage 4 significantly increased compared with GCs of passage 1. Accordingly, GCs at passage 4 were regarded as senescent GCs (GCs-S), while GCs at passage 1were considered young GCs (GCs-Y) (Figure 2B). miR-143-3p expression levels were substantially higher in GCs-S than in GCs-Y (Figure 2C). Moreover, GCs-S synthesized less E2 and P4 than GCs-Y (Figure 2D,E).

We then transfected an miR-143-3p mimic into GCs-Y to increase the exogenous miR-143-3p level while transfecting an miR-143-3p inhibitor into GCs-S. Our results indicated that the miR-143-3p mimic significantly impeded the synthesis of E2 and P4 in GCs-Y (Figure 2F,G). Conversely, transfecting GCs-S with an miR-143-3p inhibitor led to an upsurge in the concentrations of E2 and P4 (Figure 2H,I). These observations revealed that elevated miR-143-3p expression is associated with GC dysfunction.

### 2.3. Increased miR-143-3p Expression Inhibits GCs’ Proliferation and Triggers Senescence Characterized by a Distinct SASP

To investigate the potential role of miR-143-3p in GCs’ proliferation, we employed EdU incorporation assays to identify the proliferating cells. Results revealed that the miR-143-3p mimic decreased the proportion of EdU-positive cells, while inhibition of miR-143-3p had the opposite effect (Figure 3A). The cell cycle analysis demonstrated that the miR-143-3p mimic caused the GCs’ cell cycle to arrest in the G0/G1 phase, whereas the inhibition of miR-143-3p promoted their entry into the S phase (Figure 3B). Furthermore, the miR-143-3p mimic downregulated the expression of cyclin D2 (CCND2) (Figure 3C), whereas inhibition of miR-143-3p upregulated CCND2 expression (Figure 3D).

We then examined whether miR-143-3p transfection modulates SASP, which is characterized by the activation of Cxcl-1, Il-β, and Tnf-α. Our results indicated that the miR-143-3p mimic led to an increase in the levels of Cxcl-1, Il-β, and Tnf-α in GCs (Figure 3E). Conversely, the miR-143-3p inhibitor downregulated the expression of Cxcl-1, Il-β, and Tnf-α (Figure 3F). These findings suggested that overexpression of miR-143-3p induced senescence characterized by the inhibition of proliferation and a distinct SASP.

### 2.4. miR-143-3p Inhibits E2 Synthesis in GCs by Targeting Lhcgr

Our results revealed that overexpression of miR-143-3p inhibits E2 synthesis in GCs. To comprehend the mechanism of this effect, we analyzed the potential targets of miR-143-3p involved in regulating E2 synthesis, ultimately determining LHCGR as a potential target. (Figure 4A). To confirm this, we designed a luciferase reporter vector containing the mutated 3′ UTR of Lhcgr or the wild-type and co-transfected it with the miR-143-3p mimic or NC into 293T cells. Results revealed that miR-143-3p activated the 3′ UTR site of Lhcgr, suppressing luciferase activity. The suppression of Lhcgr activity was then relieved by mutating the 3′ UTR sequence (Figure 4B). Additionally, we observed a significant decrease in LHCGR expression after transfecting GCs with the miR-143-3p mimic. Conversely, inhibiting miR-143-3p increased LHCGR expression (Figure 4C,D). Similar to the effects of the miR-143-3p mimic, blocking the expression of LHCGR significantly decreased intracellular cyclic AMP (cAMP) content and inhibited E2 synthesis (Figure 4E–G). These results suggest that miR-143-3p inhibits E2 synthesis by targeting the Lhcgr.

### 2.5. miR-143-3p Inhibits GCs’ proliferation and Induces SASP Factor Upregulation by Targeting Ube2e3

To investigate the molecular mechanisms underlying the observed effects of miR-143-3p overexpression on GC proliferation and SASP factor, we conducted further investigations to identify potential targets of miR-143-3p involved in regulating proliferation, and we identified Ube2e3 as a potential target of miR-143-3p (Figure 5A). To confirm this, we constructed a luciferase reporter vector containing either the wild-type or mutated 3′ UTR of Ube2e3, which were co-transfected with either an miR-143-3p mimic or NC into 293T cells. The miR-143-3p mimic significantly reduced the luciferase activity of the wild-type Ube2e3. In contrast, there was relief in suppression when the 3′ UTR of Ube2e3 was mutated (Figure 5B), indicating that miR-143-3p directly targets Ube2e3. Furthermore, miR-143-3p overexpression significantly downregulated UBE2E3 expression, while inhibition of miR-143-3p had the opposite effect (Figure 5C). Additionally, blocking the expression of Ube2e3 significantly inhibited GCs’ proliferation (Figure 5D–F) and SASP factor (Cxcl-1, Il-β, and Tnf-α) expression (Figure 5G), similar to the effects of the miR-143-3p mimic. These results demonstrate that miR-143-3p inhibited GC proliferation and induced SASP factor expression by directly binding to the 3′ UTR of Ube2e3.

### 2.6. Overexpression of miR-143-3p Accelerated Ovarian Aging In Vivo

To further investigate the relationship between miR-143-3p expression and ovarian aging, the 8-month-old female mice were intraperitoneally injected once a week with miR-143-3p agomir to establish an in vivo model of high miR-143-3p expression, while control mice were injected with corresponding scrambled miRNAs (NC). After 4 weeks, the expression of miR-143-3p in the ovaries of the mice significantly increased (Figure 6A). Overexpression of miR-143-3p resulted in a significant decrease in serum E2 and P4 levels (Figure 6B,C). The expression of several SASP factors, including Cxcl-1, Cxcl-2, Il-6, Tnf-α, was upregulated in the ovaries of mice injected with miR-143-3p agomir (Figure 6D). Masson staining of the ovary revealed that overexpression of miR-143-3p resulted in increased ovarian fibrosis (Figure 6E). Additionally, miR-143-3p resulted in a significant increase in β-galactosidase staining-positive cells in the ovary (Figure 6F). Moreover, treatment with miR-143-3p agomir also resulted in significant downregulated levels of LHCGR and UBE2E3 in the ovaries (Figure 6G,H).

To further investigate the impact of miR-143-3p overexpression on female mice fertility, we performed breeding experiments by pairing female mice overexpressing miR-143-3p with 10-week-old wild-type male mice. The results indicated that overexpression of miR-143-3p led to a considerable reduction in litter size. Compared to normal mice and those mice treated with NC, which had an average litter size of seven pups per litter, miR-143-3p overexpression resulted in a decreased average of one pup per litter (Figure 6I,J). 

These phenotypes observed in the ovaries of mice overexpressing miR-143-3p closely resembled those seen in 15-month-old female mice. This suggested that miR-143-3p overexpression induced ovarian ageing, dysfunction and ultimately reduced fertility in mice.

## 3. Discussion

In this study, we found that ovarian dysfunction in aged mice is correlated with elevated levels of miR-143-3p in ovaries. Our results revealed that miR-143-3p plays a dual role in regulating GCs’ senescence. On one hand, miR-143-3p downregulated the expression of LHCGR, which in turn reduced cAMP production and limited steroid hormone synthesis, ultimately resulting in decreased levels of E2. On the other hand, miR-143-3p directly targets UBE2E3, leading to suppressed GC proliferation and inducing SASP factors. Collectively, these effects of miR-143-3p contribute to the age-related decline in ovarian hormonal function and GCs’ senescence (Figure 7).

miR-143-3p has been identified as playing an essential role in tumor development, progression, and metastasis. For example, in brain metastasis of lung cancer, miR-143-3p has been shown to promote lung cancer metastasis and angiogenesis by targeting vasohibin-1 (VASH1) [34]. Furthermore, elevated circulating levels of miR-143-3p were observed in patients with acute ischemic stroke [35]. In the female reproductive system, miR-143-3p has been found to be highly expressed in follicular-fluid-derived exosomes of patients with polycystic ovary syndrome [36]. miR-143-3p not only antagonistically regulated glycolytic-mediated follicular dysplasia of GCs by targeting hexokinase 2 (HK2) [36], but also promoted GCs’ apoptosis by targeting BMPR1A [37]. These studies demonstrated that an inverse correlation between miR-143-3p levels and cell viability, with high levels of miR-143-3p being associated with ovarian dysfunction. In our study, we observed elevated miR-143-3p expression in the ovaries of aged mice compared to their younger counterparts. These aged mice displayed decreased fertility, lower steroid hormone levels, and increased ovarian expression of miR-143-3p, suggesting a potential relationship between age-related declines in reproductive function and miR-143-3p expression. 

Ovarian GCs are responsible for synthesizing and secreting estrogen in response to FSH from the pituitary gland [38]. Dysfunction of GCs can lead to irregular or absent menstrual cycles, infertility, or hormonal imbalances [39]. We hypothesized that miR-143-3p plays a role in the age-related decline in GCs’ function. To investigate the role of miR-143-3p in GCs, we isolated primary GCs. These primary GCs can be passaged a limited number of times when cultured in vitro before entering a phase of irreversible growth arrest, known as replicative senescence, which is thought to reflect cellular aging in vivo [40,41]. We observed a significant and progressive increase in miR-143-3p expression in GCs with each cell passage. Subsequently, primary GCs were transfected with an miR-143-3p mimic, resulting in a significant reduction in the synthesis of both E2 and P4, cell proliferation arrest, and increased levels of SASP. The miR-143-3p inhibited E2 synthesis in GCs by targeting LHCGR, which plays a critical role in ovulation and pregnancy by binding to LH and promoting E2 and progesterone production [42]. In GCs, LH stimulates the expression of LHCGR, which leads to the activation of intracellular signaling pathways cAMP pathway. This, in turn, stimulates the synthesis and secretion of E2 by the GCs. LHCGR also regulates the secretion of progesterone, which is crucial for the maintenance of pregnancy and the preparation of the uterus for implantation [43]. Dysfunction of LHCGR in GCs can lead to ovarian insufficiency, impaired ovulation, infertility, and, in some cases, ovarian tumors [16,44]. Our finding also demonstrated that knocking down of LHCGR resulted in reduced E2 synthesis in GCs. These results reinforce the significance of miR-143-3p as a potential therapeutic target for hormonal imbalances of GC.

Additionally, increased expression of SASP markers in the ovaries led to premature ovarian failure, suggesting that SASP may contribute to ovarian ageing [45]. Moreover, inhibiting GC proliferation may interfere with proper follicular development and resulted in a decline in ovarian function [46]. Our findings revealed that miR-143-3p overexpression inhibited GC proliferation and induced SASP factors by targeting UBE2E3. UBE2E3 is a highly conserved metazoan ubiquitin conjugating enzyme that has been found to be involved in cell proliferation and senescence [47]. Knocking down of UBE2E3 in retinal pigment epithelial cells (RPE) resulted in a robust increase in p27^Kip1^ and led to cell cycle exit [47]. Further studies revealed that depletion of UBE2E3 induced cellular senescence, which could be partially suppressed by co-depletion of p53 or its cognate target genes, p21^CIP1/WAF1^, or by co-depletion of the tumor suppressor gene p16^INK4a^. The mechanism by which UBE2E3 mediated cellular senescence was that UBE2E3 deficiency increased the sensitivity of mitochondria to toxins, leading to collapse of the mitochondrial network and an increase in basal autophagy/ mitophagy [48]. Moreover, Mulan E3 ubiquitin ligase and UBE2E3 formed Mulan–Ube2e3 complexes, which then recruited GABARAP (GABAA receptor-associated protein) to participate in the process of mitosis [49]. Studies have found that UBE2E3 regulated cellular senescence in bone marrow mesenchymal stem cells (BMSCs) during aging and that this regulatory effect was achieved by modulating the function of the nuclear factor erythroid 2-related factor (Nrf2) [50]. Indeed, another study found that UBE2E3, together with its nuclear import receptor importin 11 (Imp-11), regulated Nrf2 distribution and activity in cells. UBE2E3 bound to Imp-11 in the cytoplasm and the complex translocated to the nucleus, where it dissociated and was released. UBE2E3 bound and promoted nuclear Nrf2 activity, whereas Imp-11 functions to limit the premature extraction of transcription factors from a subset of the promoters of target genes by KEAP1, a major repressor of Nrf2. Deletion of UBE2E3 resulted in the redistribution of Nrf2 from the nucleus to the cytoplasm and a decrease in the transcriptional activity of Nrf2, ultimately driving proliferating cells into senescence [51]. This suggested that UBE2E3 may play a role in regulating cellular processes that contribute to maintaining mitochondrial health and preventing senescence. Consistent with previous studies, our study found that knockdown of UBE2E3 in GCs resulted in blocking of proliferation and inducing SASP factors. Moreover, miR-143-3p overexpression in vivo significantly accelerated ovarian aging. These results implied that increased miR-143-3p expression could contribute to ovarian aging and fertility decline by affecting GC proliferation and inducing senescence by inhibiting UBE2E3 expression. 

However, aging is a highly complex biological process. The molecular mechanisms that regulate aging at the cellular level are largely unknown [52]. The regulation of ovarian aging is a complex process involving various molecular pathways, and miRNAs have emerged as crucial regulators in this context. The current strategies for prevention and treatment of ovarian aging are limited. Hormone replacement therapy (HRT) and assisted reproductive technologies (ART) are the commonly employed treatments [53]. However, HRT not only magnifies the risk of hormone-related cancers, such as breast and ovarian cancer, but also heightens the risk of cardiovascular diseases [54,55]. Moreover, due to the substantial reduction or depletion of ovarian reserves in aging ovaries, assisted reproductive techniques can only achieve limited effectiveness [56]. Our findings demonstrated that miR-143-3p directly bound to the 3’ UTR of the UBE2E3 gene, leading to the inhibition of UBE2E3 expression and impacting the process of ovarian cell senescence. As such, interventions aimed at modulating miR-143-3p levels or activity could potentially alter the senescence-associated changes in ovarian cells and improve overall ovarian function. For example, pharmaceutical or gene therapy approaches may be designed to modulate miR-143-3p expression or activity in order to restore or enhance the expression of UBE2E3 and counteract ovarian senescence. These interventions could potentially rejuvenate ovarian tissues, improve oocyte quality, and increase fertility potential in women of advanced reproductive age. However, this requires more rigorous and detailed studies on the upstream regulatory mechanisms and specific molecular mechanisms of downstream targets of miR-143-3p. Further studies are needed to fully understand the broad implications of miR-143-3p regulation and its potential therapeutic applications for ovarian aging. For instance, the effects of interventions targeting miR-143-3p in the aged mice should be examined. 

In summary, we identified that miR-143-3p plays a significant role in ovarian aging by inhibiting E2 and P4 synthesis, hindering GC proliferation, and inducing senescence. Targeting miR-143-3p could serve as a potential therapeutic approach to treat fertility decline and ovarian aging in women.

## 4. Materials and Methods

### 4.1. Animals

C57BL/6J female mice (3 to 15 months old) and KM female mice (3 weeks old) were purchased from the Experiment Animal Center of Guangdong Province (Guangzhou, China), and maintained under 12 h light/dark cycle at controlled temperature (24 ± 2 °C) with relative humidity of 50–60%. Standard rodent diet and drinking water were freely accessible. Prior to the study, the C57BL/6J female mice were acclimatized for a minimum of one week. In the first part of our study, to investigate the relationship between age and miR-143-3p in mice, female mice were divided by age into one of the following groups (*n* = 6): young group (4 months old, 23 g average body weight), middle-aged group (8 months old, 28 g average body weight), and an old group (15 months old, 37 g average body weight). All animal experiments were conducted in accordance with the National Institute of Health Guidelines for the care and use of animals and approved by the Institutional Animal Care and Use Committee of Jinan University.

### 4.2. GCs Isolation and Culture

GCs were isolated from 3-week-old female KM mice. Briefly, the mice were sacrificed, and their ovaries were rapidly isolated under aseptic conditions by using a pair of scissors and pointed forceps. To release GCs, a 5-gauge needle was used to puncture each ovary. Then, the cell suspension was filtered through a 100-mesh cell sieve. The GCs were collected by concentrating at 250 g for 5 min and resuspended in DMEM-F/12 (Gibco, Grand Island, NY, USA) with 15% fetal bovine serum (FBS, Life Technologies, Carlsbad, CA, USA).

### 4.3. Protein Extraction and Western Blot

Total protein of the cultured cells and mouse ovaries were harvested with RIPA lysis buffer (Fude Bio, Hangzhou, China). The protein concentration was determined using a BCA kit. The protein components were separated using SDS-PAGE and transferred to polyvinylidene fluoride membranes. After blocking with 5% skim milk for 1 h, the membranes were incubated with β-actin, LHCGR, UBE2E3, FSHR, and CCND2 antibodies overnight at 4 °C and then incubated with a secondary antibody labeled with horseradish peroxidase for 1 h at room temperature. Finally, enhanced chemiluminescence (ECL) reagent was used to detect and analyze the immunoreactive bands. Antibody information is listed in Table 1. 

### 4.4. RNA Extraction and Quantitative Real-Time Polymerase Chain Reaction (qRT-PCR)

Trizol reagent (Life, Carlsbad, CA, USA) was used to extract total RNA from all samples. The extracted total RNA was dissolved in RNase-free water and then reverse transcribed to cDNA using the PrimeScript RT Reagent Kit (Takara, RR047A, Otsu, Japan). The mRNA expression levels of target gene were assessed by RT real-time PCR with SYBRR Premix Ex Tap™ (TaKaRa, Otsu, Japan). Then, qRT-PCR was performed using BioRad CFX manager system (BioRad, Hercules, CA, USA) at an initial activation 95 °C for 10 min, followed by 40 cycles of 95 °C for 10 s, 60 °C for 20 s, and 72 °C for 10 s. The level of gene expression was normalized to the reference gene GAPDH. Data analysis was carried out using the 2^−ΔΔCt^ method. The primers used for qRT-PCR are listed in Table 2. 

### 4.5. Oligonucleotide Transfection

miR-143-3p mimic, mimic negative control, miR-143-3p inhibitor and inhibitor negative control oligonucleotides were chemically synthesized by GenePharma (Shanghai, China). GCs were seeded in 6-well plates at a density of 4 × 10^5^ cell/well for 24 h and grown to a confluence of 70–80%. Oligonucleotides were then transfected into the GCs at a final concentration of 60 pM using RNAiMAX (Invitrogen, Carlsbad, CA, USA) for 24 h or 72 h. Then, the GCs and their supernatant were collected for further analysis.

### 4.6. Hormone Assay

The concentrations of follicle-stimulating hormone, luteinizing hormone, and estradiol were measured according to the manufacturer’s guide using ELISA kits (Elabscience, Wuhan, China). The concentration of progesterone was measured with Iodine [^125^I] Radioimmunoassay Kit (Bejing North Institute of Biotechnology, Beijing, China).

### 4.7. Intraperitoneal Injection of miR-143-3p Agomir

miR-143-3p agomir and negative control were purchased from Ribobio (RiboBio, Guangzhou, China). Nucleic acid drugs were diluted with normal saline at a concentration of 0.6 nmol/100 μL. Each 8-month-old female mouse was injected intraperitoneally with 100 uL of the above solution once a week for 4 weeks.

### 4.8. Luciferase Reporter Assay

miR-143-3p was predicted to bind to the CDS region of *Lhcgr* mRNA and the CDS region of *Ube2e3* mRNA using the Targetscan database and miRwalk database. Luciferase reporter assay was performed using dual glo luciferase assay system kit (Promega, Madison, WI, USA) according to the manufacturer’s instructions. A luciferase reporter plasmid (psi-check 2 containing wild-type, mutant *Lhcgr* CDS region or mutant *Ube2e3* CDS region) was constructed and co-transfected with miR-143-3p mimic or miR-143-3p NC, respectively, in HEK293T cells. After transfection for 48 h, cells were analyzed with dual glo Luciferase Assay system kit. Firefly luciferase (F-luc) was used to normalize Renilla luciferase (R-luc) activity to evaluate reporter translation efficiency. 

### 4.9. Fluorescence In Situ Hybridization (FISH)

Initially, the paraffin sections of the ovaries were incubated for 2 h at 56 °C, deparaffinized in xylene, and dehydrated with ethanol gradients: 100%, 95%, 75%, 50% and 0% ethanol concentrations for 5 min each. Sections were washed with PBS and placed in 1× sodium citrate (SCC) buffer at 75 °C for 10 min, turned to 45 °C for 15 min, and cooled at room temperature for 30 min. Next, after being washed with PBS for 5 min, the sections were soaked in 3% hydrogen peroxide solution. Then, we washed the sections with PBS for 5 min, shook off excess water from the sections, and added probe (the probe sequence is listed in Table 2) to selected hybridization regions at 37 °C for 24 h. Posthybridization washes were performed with 2× SSC, followed by wash in 2× SSC for 2 min and dehydration in ethanol series. Chromatin was counterstained with DAPI and mounted with Mowiol. At last, the results were analyzed using fluorescence microscopy. 

### 4.10. EdU Assay

GCs’ proliferation was assessed using the 5-ethyl-20-deoxyuridine (EDU) incorporation method. The EDU assay procedure was performed according to the instructions of the kit manufacturer (Keygen, Nanjing, China). GCs were incubated with 10 μM EdU medium for 48 h. Then, the cells were washed with PBS and stained with DAPI to visualize EDU-positive cells.

### 4.11. SPIDER-β-Gal Staining

GCs were fixed with 4% PFA, and then incubated with SPIDER-β-Gal in PBS for 30 min in 37 °C. The nucleuses were stained with DAPI. Following the wash with PBS, GCs were observed under fluorescence microscopy.

### 4.12. Mating Experiment

For the detection of fertility in female mice at different ages, 4-month-old, 8-month-old, and 15-month-oldsexually mature C57BL/6J female mice and 3-month-oldmale mice were caged 1:1 for 14 days, after which the male mice were removed to check the fertility and litter size of the female mice. For the detection of fertility in female mice injected with miR-143-3p agomir and negative control, after the 8-month-old C57BL/6J female mice completed intraperitoneal injection, they were caged with 3-month-old male C57BL/6J mice for 14 days, after which the males were removed, and the female mice were tested for litter size. 

### 4.13. Statistical Analysis

The statistical analyses were performed with GraphPad Prism 8.0 software. All data were shown to fit a normal distribution by using the Shapiro–Wilk test. The results are shown as the mean values ± 1 SD. Unpaired student’s t test was used to analyze the data between two groups. One-way ANOVA was used to determine significant intergroup differences. *p*-values less than 0.05 were considered significant and were marked as follows: * *p* < 0.05, ** *p* < 0.01, *** *p* < 0.001 and **** *p* < 0.0001.

## 5. Conclusions

This study revealed that miR-143-3p was highly expressed in senescent ovarian GCs and was correlated with age-related decline in female mice fertility. Overexpression of miR-143-3p in GCs could hinder their proliferation and induce senescence-associated secretory phenotype (SASP), as well as impede steroid hormone synthesis by targeting *Ube2e3* and *Lhcgr*. These findings suggest that miR-143-3p plays a significant role in senescence and steroid hormone synthesis in GCs, indicating its potential as a therapeutic target for interventions in the ovarian aging process. 

## Figures and Tables

**Figure 1 ijms-24-12560-f001:**
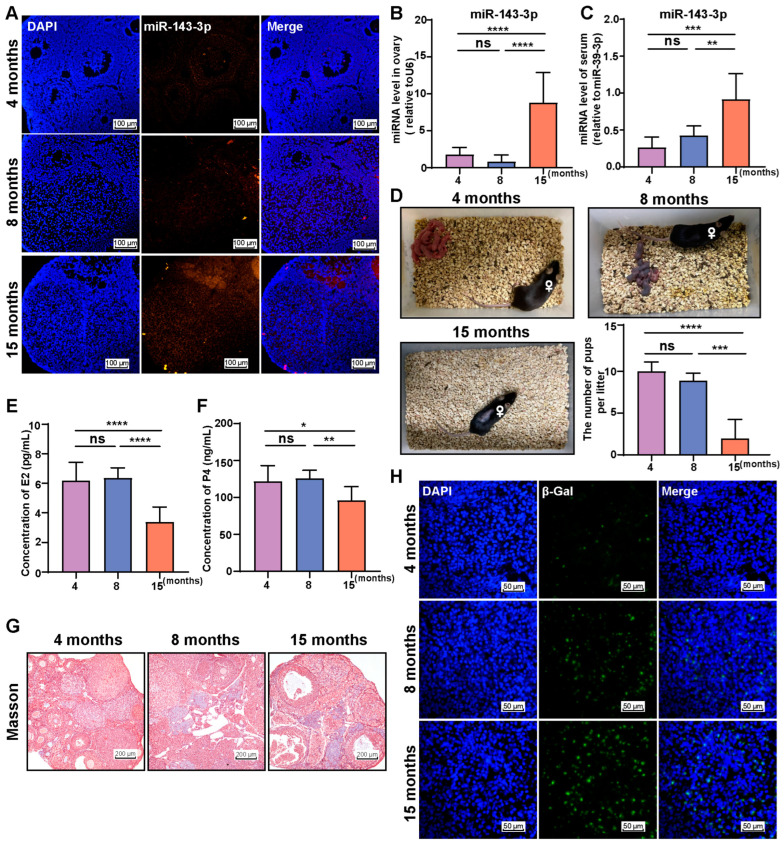
miR-143-3p is highly expressed in the ovaries of aged mice. (**A**) Representative fluorescence in situ hybridization images of miR-143-3p expression in ovarian tissue from young and aged mice. The scale bar = 100 μm. (**B**) Expression level of miR-143-3p in ovaries from mice of different ages. (**C**) Expression level of miR-143-3p in serum from female mice of different ages. (**D**) Fertility analysis of mice of different ages. (**E**) Serum E2 concentration in mice of different ages. (**F**) Serum P4 concentration in mice of different ages. (**G**) Masson staining of the ovaries from mice at different ages. The scale bar = 200 μm. (**H**) Representative images of β-galactosidase staining in ovarian tissue from mice of different ages. The scale bar = 50 μm. All data are presented as the mean ± SD from at least three independent experiments. * *p* < 0.5, ** *p* < 0.01, *** *p* < 0.001, **** *p* < 0.0001, ns = not significant.

**Figure 2 ijms-24-12560-f002:**
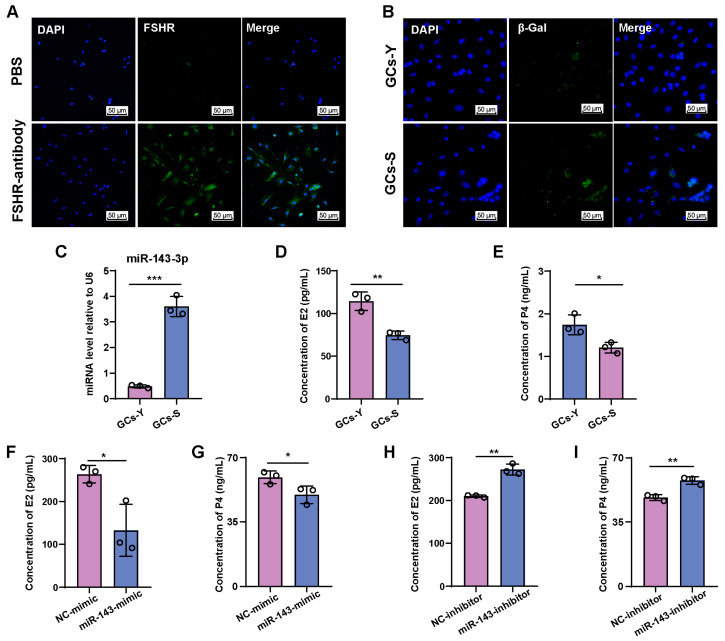
miR-143-3p inhibits E2 and P4 synthesis in GCs. (**A**) FSHR immunofluorescence of GCs isolated from ovaries. The scale bar = 50 μm. Green, FSHR positive. Blue, DAPI. (**B**) β-galactosidase staining in young and senescent GCs. The scale bar = 50 μm. Green: β-galactosidase positive. (**C**) Expression level of miR-143-3p in young and senescent GCs. (**D**,**E**) E2 and P4 concentrations in culture supernatant from young and senescent GCs. GCs-Y = young GCs, GCs-S = senescent GCs. (**F**,**G**) E2 and P4 concentrations in culture supernatant from GCs transfected with an miR-143-3p mimic. (**H**,**I**) E2 and P4 concentrations in culture supernatant from GCs transfected with an miR-143-3p inhibitor. NC-mimic = mimic negative control, NC-inhibitor = inhibitor negative control. Data are presented as the mean ± SD from at least three independent experiments. * *p* < 0.5, ** *p* < 0.01, *** *p* < 0.001.

**Figure 3 ijms-24-12560-f003:**
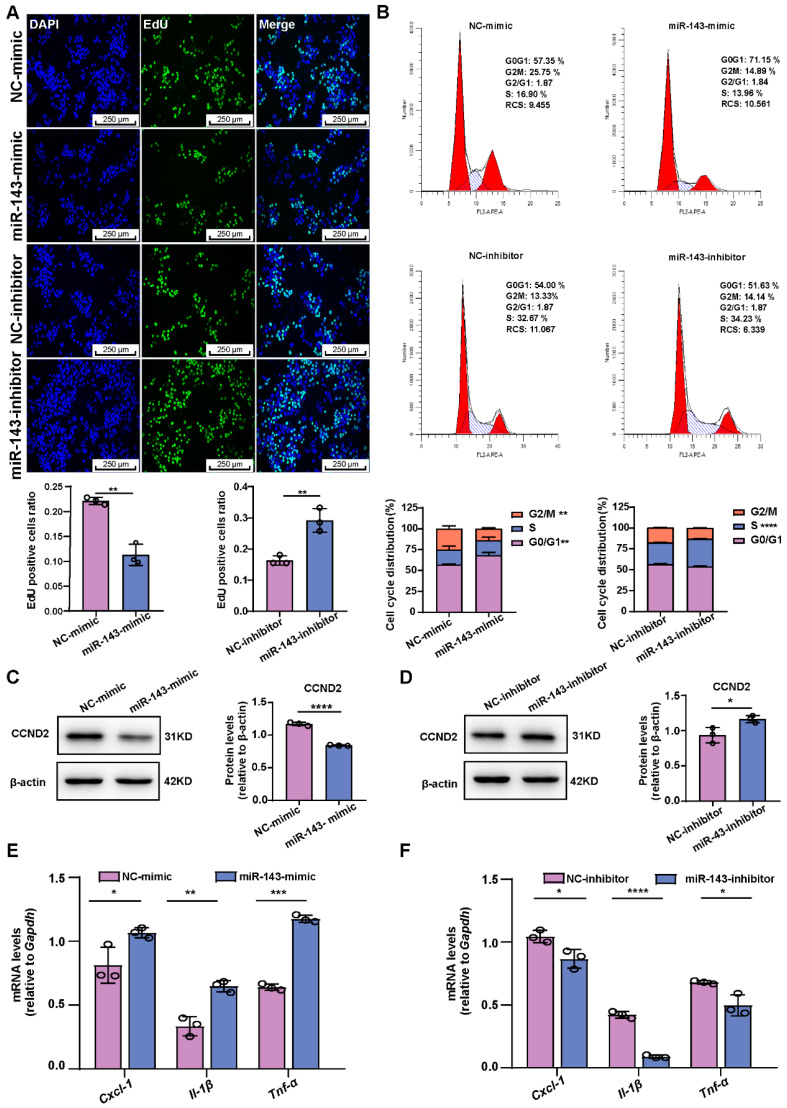
miR-143-3p inhibits GCs’ proliferation and induces senescence characterized by a distinct SASP. (**A**) EdU assays showed the effect of the miR-143-3p mimic and inhibitor on GCs’ proliferation. The scale bar = 250 μm. Green, EdU positive. Blue, DAPI. (**B**) Flow cytometry results showed the impact of miR-143-3p on the cell cycle of GCs transfected with an miR-143-3p mimic, miR-143-3p inhibitor, or the respective NC. (**C**,**D**) Protein level of CCND2 in GCs transfected with miR-143-3p mimic and an miR-143-3p mimic, miR-143-3p inhibitor, or the respective NC. (**E**,**F**) qRT-PCR analysis of the indicated SASP factors in GCs transfected with an miR-143-3p mimic, miR-143-3p inhibitor, or the respective NC. NC-mimic = mimic negative control, NC-inhibitor = inhibitor negative control. Data are presented as the mean ± SD from at least three independent experiments. * *p* < 0.5, ** *p* < 0.01, *** *p* < 0.001, **** *p* < 0.0001.

**Figure 4 ijms-24-12560-f004:**
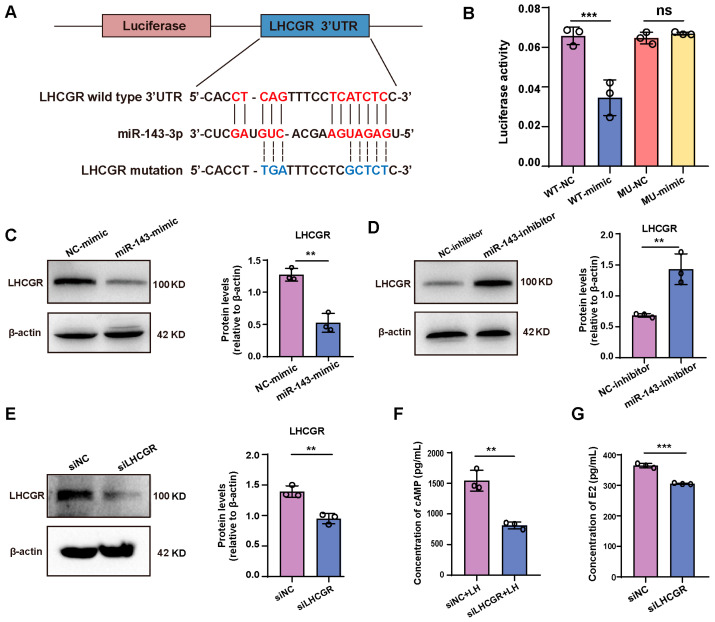
miR-143-3p inhibits E2 synthesis in GCs by targeting *Lhcgr.* (**A**) miR-143-3p was predicted to bind to the 3’ UTR region of *Lhcgr* mRNA. (**B**) A luciferase reporter plasmid containing either the wild-type or mutant UTR region of *Lhcgr* was constructed and co-transfected with the miR-143-3p mimic or miR-143-3p NC, respectively, in 293T cells. The relative Renilla luciferase activity was normalized to the firefly luciferase activity. (**C**,**D**) Protein levels of LHCGR in GCs transfected with miR-143-3p mimic, miR-143-3p inhibitor, or the respective NC. (**E**) Protein level of LHCGR in GCs transfected with siNC and siLHCGR. (**F**) Intracellular cAMP concentration in GCs transfected with siNC and siLHCGR. (**G**) E2 concentration in culture supernatant from GCs transfected with siNC and siLHCGR. siNC = negative control siRNA, siLHCGR = *Lhcgr* siRNA. All data are presented as the mean ± SD from at least three independent experiments. ** *p* < 0.01, *** *p* < 0.001, ns = not significant.

**Figure 5 ijms-24-12560-f005:**
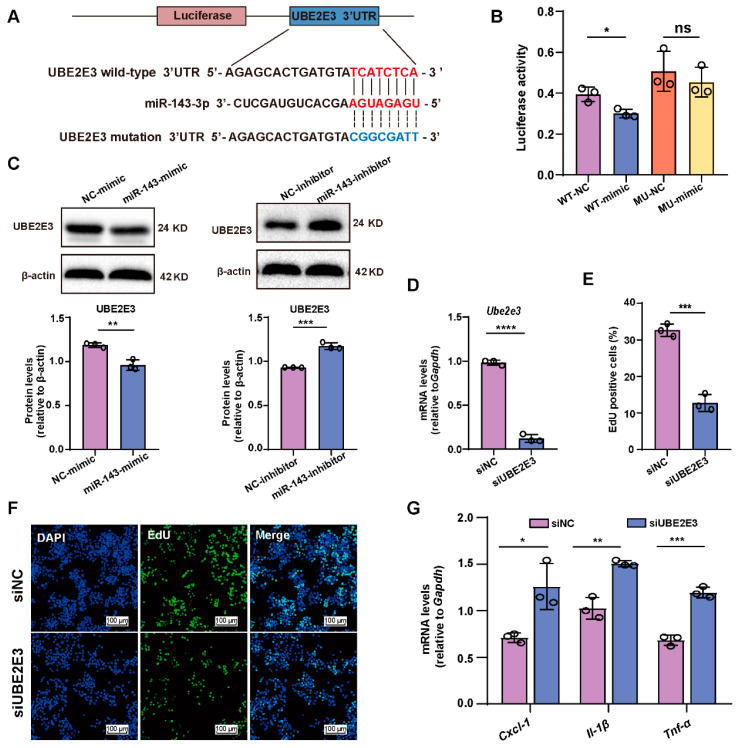
miR-143-3p inhibits GCs’ proliferation and induces SASP factor upregulation by targeting *Ube2e3*. (**A**) miR-143-3p was predicted to bind to the UTR region of *Ube2e3* mRNA. (**B**) A luciferase reporter plasmid containing either wild-type or mutant *Ube2e3* UTR region was constructed and co-transfected with miR-143-3p mimic or miR-143-3p NC, respectively in 293T cells. The relative Renilla luciferase activity was normalized to the firefly luciferase activity. (**C**) Protein level of UBE2E3 in GCs transfected with miR-143-3p mimic, miR-143-3p inhibitor, or the respective NC. (**D**) qRT-PCR analysis of the interference effect of siUBE2E3. (**E**,**F**) EdU assays showed the effect on cell proliferation following *Ube2e3* knockdown in GCs. siNC = negative control siRNA, siUBE3E3 = *Ube2e3* siRNA. Green, EdU positive. Blue, DAPI. The scale bar = 100 μm. (**G**) qRT-PCR analysis of the SASP factors in GCs transfected with siNC or siUBE2E3. All data are presented as the mean ± SD from at least three independent experiments. * *p* < 0.5, ** *p* < 0.01, *** *p* < 0.001, **** *p* < 0.0001, ns = not significant.

**Figure 6 ijms-24-12560-f006:**
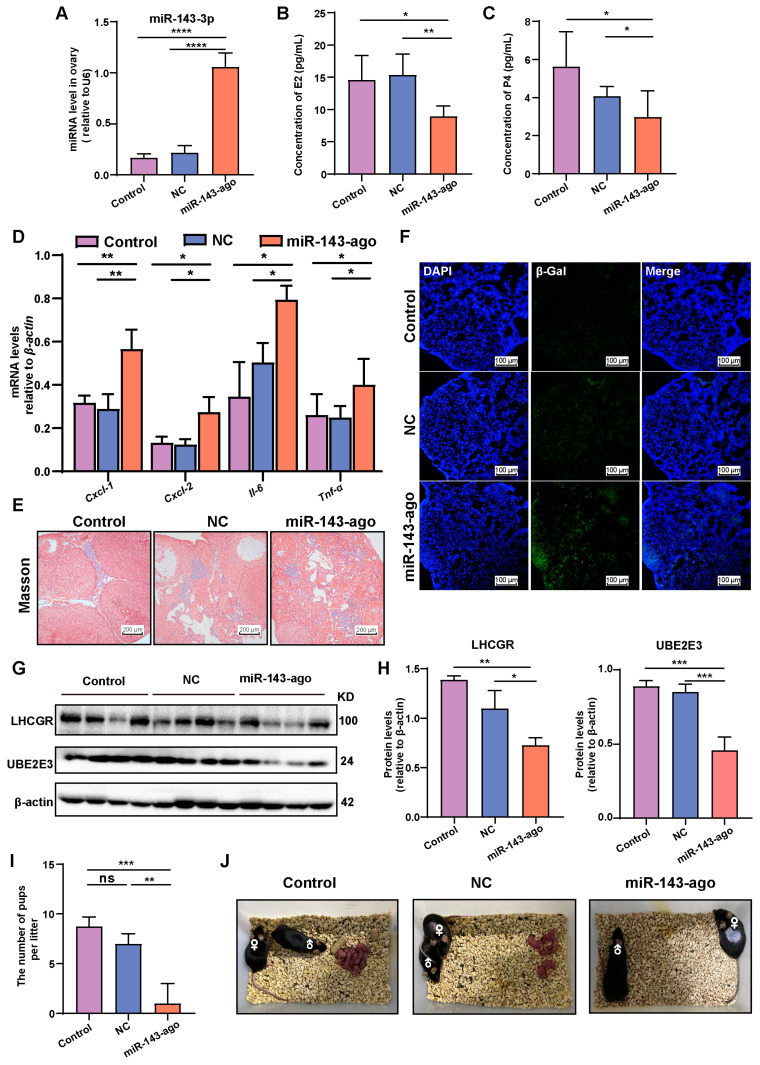
Overexpression of miR-143-3p accelerated ovarian aging in vivo. (**A**) Efficiency of miR-143-3p agomir intraperitoneal injection in mice was confirmed using qRT-PCR. miR-143-ago represented miR-143-3p agomir. (**B**,**C**) Serum E2 and P4 concentrations of control mice and mice injected with miR-143-3p agomir and negative control. (**D**) qRT-PCR analysis of the indicated SASP factors in ovaries from mice after injection with miR-143-3p agomir. (**E**) Masson staining of the ovaries from mice injected with miR-143-3p agomir and negative control. The scale bar = 200 μm. (**F**) β-galactosidase staining of the ovaries from mice after injection with miR-143-3p agomir. The scale bar = 100 μm. Green, β-galactosidase positve. Blue, DAPI. (**G**,**H**) The expressions of LHCGR and UBE2E3 in mice after injection with miR-143-3p agomir. (**I**,**J**) The litter size of control mice and mice injected with miR-143-3p agomir and negative control. All data are presented as the mean ± SD from at least three independent experiments. * *p* < 0.5, ** *p* < 0.01, *** *p* < 0.001, **** *p*< 0.0001, ns = not significant.

**Figure 7 ijms-24-12560-f007:**
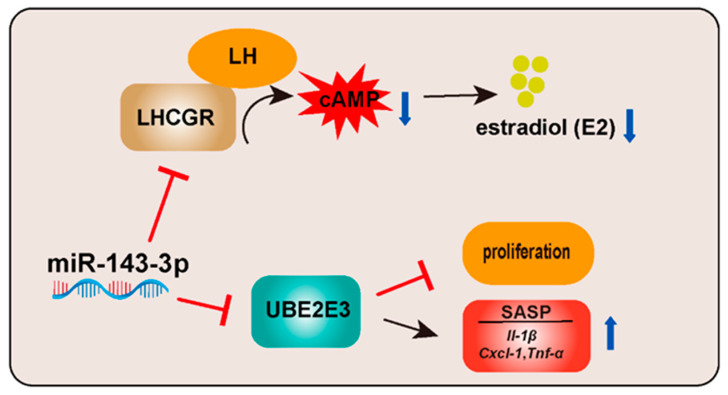
Schematic illustration of how miR-143-3p contributes to the age-related decline in ovarian hormonal function and GC senescence.

**Table 1 ijms-24-12560-t001:** Antibodies used in the WB.

Name	Manufacturers	Usage and Dosage
UBE2E3	Proteintech	WB (1:1000)
LHCGR	Proteintech	WB (1:1000)
CCND2	CST	WB (1:1000)
β-actin	Fude bio	WB (1:5000)

**Table 2 ijms-24-12560-t002:** Primers for qRT-PCR.

Primers	Sence Primer (5’-3’)	Anti-Sence Primer
*Cxcl-1*	TGCACCCAAACCGAAGTCAT	CTCCGTTACTTGGGGACACC
*Cxcl-2*	TCATAGCCACTCTCAAGGGC	TCAGGTACGATCCAGGCTTC
*Il-1β*	GCCACCTTTTGACAGTGATGAG	GACAGCCCAGGTCAAAGGTT
*Il-6*	CACTTCACAAGTCGGAGGCT	CTGCAAGTGCATCATCGTTGT
*Il-8*	CTTTGTCCATTCCCACTTCTGA	TCCCTAACGGTTGCCTTTGTAT
*Tnf-α*	ATGTCTCAGCCTCTTCTCATTC	GCTTGTCACTCGAATTTTGAGA
Fluorescence In Situ Hybridization		Sence primer (5’-3’)
miR-143-3p		GAGCTACAGTGCTTCATCTCA

## Data Availability

Data are available upon request.

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
