# Peer review of "miR-143-3p Promotes Ovarian Granulosa Cell Senescence and Inhibits Estradiol Synthesis by Targeting UBE2E3 and LHCGR"

_ijms, 2023, doi:10.3390/ijms241612560_

Round 1

Reviewer 1 Report

Overall, the manuscript presents an interesting study investigating the role of miR-143-3p in ovarian granulosa cell (GC) senescence and hormone synthesis. However, The introduction could be improved by providing more context on the biological pathways involved in ovarian aging, particularly focusing on the roles of granulosa cells (GCs) and miR-143-3p. This would allow readers to better understand the significance of the study.

The authors should provide more information on the potential mechanisms involved in miR-143-3p regulation of GC senescence. For example, how does miR-143-3p directly target UBE2E3 and what is the downstream effect of this targeting?

The authors should also consider discussing the implications of their findings in the context of potential therapeutic targets for ovarian aging. How might interventions targeting miR-143-3p improve ovarian function and fertility in aging women?

To validate the direct targeting of UBE2E3 and LHCGR by miR-143-3p, the authors could perform luciferase reporter assays to confirm binding specificity and efficiency. Additionally, they could consider using CRISPR/Cas9 gene editing to generate UBE2E3 or LHCGR knockout GCs to confirm the role of these targets in GC senescence and hormone synthesis.

To explore the potential therapeutic implications of their findings, the authors could test the effects of interventions targeting miR-143-3p in animal models of ovarian aging. For example, they could use miRNA inhibitors or other small molecules to reduce miR-143-3p expression and evaluate the effects on ovarian function and fertility.

To better understand the molecular mechanisms involved in miR-143-3p regulation of GC senescence, the authors could perform RNA-sequencing (RNA-seq) and proteomic analyses to identify downstream targets and signaling pathways affected by miR-143-3p. Also, The authors can do the IPA analysis as well. 

Author Response

Dear reviewers,

Thank you for giving us an opportunity to revise our manuscript, we would like to express our gratitude to the editor and reviewers for their positive and constructive comments and suggestions on our manuscript entitled “miR-143-3p promotes ovarian granulosa cell senescence and inhibits estradiol synthesis by targeting UBE2E3 and LHCGR” (manuscript ID: ijms-2515667). These comments are all valuable and helpful for revising and improving our manuscript. We carefully studied all comments and made conscientious correction. And we have provided a point-by-point response to the comments and marked the revised portions of the paper for easy identification. The main corrections made to the paper and our responses to the reviewers' comments are as follows:

Responds to the reviewer’s comments:

Reviewer #1: Overall, the manuscript presents an interesting study investigating the role of miR-143-3p in ovarian granulosa cell (GC) senescence and hormone synthesis.

Comment 1. The introduction could be improved by providing more context on the biological pathways involved in ovarian aging, particularly focusing on the roles of granulosa cells (GCs) and miR-143-3p. This would allow readers to better understand the significance of the study.

Response: We appreciate for your valuable suggestions for improving the quality of our work. We have added some introductions about biological pathways involved in ovarian aging.

Revised manuscript, line 67-77

“As age increases, the number of senescent GCs in the ovary also increases. Aging GCs exhibit dysregulated signaling pathways involved in various cellular processes. For instance, the accumulation of a significant amount of reactive oxygen species (ROS) in GCs inhibited Wnt16 signaling, leading to increased β-catenin activity and subsequently promoting GC aging [20]. Decreased activity of the PI3K/Akt pathway inhibited GC proliferation and accelerated GC apoptosis [21]. Studies have shown that as women aged, the structure and function of mitochondria within GCs undergo changes, including a reduction in the number of mitochondria, accumulation of mitochondrial DNA damage, and a decline in mitochondrial respiratory chain activity. These changes may result in compromised follicular development and diminished oocyte quality [22].”

Revised manuscript, line 98-100

“The intronic microRNA let-7g targets and downregulates transforming growth factor-β type 1 receptor (TGFBR1) to inhibit the TGF-β/SMAD signaling pathway thereby promoting GCs apoptosis [31].”

Revised manuscript, line 102-105

“The miR-143-3p has been reported to be involved in regulating tumor development and cardiovascular diseases [32, 33]. In GCs, miR-143-3p targeted and downregulated FSHR expression during follicular atresia [14]. However, the role of miR-143-3p in GC senescence remains poorly understood.”

Comment 2. The authors should provide more information on the potential mechanisms involved in miR-143-3p regulation of GC senescence. For example, how does miR-143-3p directly target UBE2E3 and what is the downstream effect of this targeting?

Response: We agree with the reviewer that it is important to reveal the potential mechanisms involved in miR-143-3p regulation of GC senescence.

In our research, we have found that miR-143-3p could directly bind to the 3' UTR of the LHCGR and UBE2E3 gene and suppressed its expression. This was achieved through complementary sequence matching, where a partial sequence of miR-143-3p matched with the 3' UTR sequence of LHCGR and UBE2E3.

Regarding the downstream effects of miR-143-3p targeting UBE2E3, our study findings suggested that this regulatory mechanism led to the inhibition of UBE2E3, thereby impacting the process of GC cell senescence. Studies showed that UBE2E3 is involved in cell proliferation and senescence. UBE2E3 deficiency increased the sensitivity of mitochondria to toxins, leading to col-lapse of the mitochondrial network and an increase in basal autophagy/mitophagy. On the other hand, deletion of UBE2E3 resulted in the redistribution of Nrf2 from the nucleus to the cytoplasm, a decrease in the transcriptional activity of Nrf2 and ultimately drive proliferating cells into senescence.

And in the revised manuscript we discussed this issue in the “Discussion section”. Please check line 343-366. And inspired by the comments of the reviewer, we believe that further investigation into the specific mechanisms by which UBE2E3 affects GC senescence will provide a more comprehensive understanding of this process.

Discussion section:

“The UBE2E3 is a highly conserved metazoan ubiquitin conjugating enzyme that has been found to be involved in cell proliferation and senescence [47]. Knocking down of UBE2E3 in retinal pigment epithelial cells (RPE) resulted a robust increase in p27 Kip1 and led to cell cycle exit [47]. Further studies revealed that depletion of UBE2E3 induced cellular senescence, which could be partially suppressed by co-depletion of p53 or its cognate target genes, p21CIP1/WAF1, or by co-depletion of the tumor suppressor gene p16INK4a. The mechanism by which UBE2E3 mediated cellular senescence was that UBE2E3 deficiency increased the sensitivity of mitochondria to toxins, leading to collapse of the mitochondrial network and an increase in basal autophagy/ mitophagy [48]. Moreover, Mulan E3 ubiquitin ligase and UBE2E3 formed Mulan–Ube2e3 complexes, which then recruited GABARAP (GABAA receptor-associated protein) to participate in the process of mitosis [49]. Studies have found that UBE2E3 regulated cellular senescence in bone marrow mesenchymal stem cells (BMSCs) during the aging and that this regulatory effect was achieved by modulating the function of the nuclear factor erythroid 2-related factor (Nrf2) [50]. Indeed, another study found that UBE2E3, together with its nuclear import receptor importin 11 (Imp-11), regulated Nrf2 distribution and activity in cells. UBE2E3 bound to Imp-11 in the cytoplasm and the complex translocated to the nucleus where it dissociated and was released. UBE2E3 bound and promoted nuclear Nrf2 activity, whereas Imp-11 functions to limit the premature extraction of transcription factors from a subset of the promoters of target genes by KEAP1, a major repressor of Nrf2. Deletion of UBE2E3 resulted in the redistribution of Nrf2 from the nucleus to the cytoplasm, a decrease in the transcriptional activity of Nrf2 and ultimately drive proliferating cells into senescence [51].”

Comment 3. The authors should also consider discussing the implications of their findings in the context of potential therapeutic targets for ovarian aging. How might interventions targeting miR-143-3p improve ovarian function and fertility in aging women?

Response: We appreciate the reviewer's suggestion to discuss the implications of our findings in the context of potential therapeutic targets for ovarian aging. We have discussed this issue in the “Discussion section”. Please check line 374-396.

“The regulation of ovarian aging is a complex process involving various molecular pathways, and miRNAs have emerged as crucial regulators in this context. The current strategies for prevention and treatment of ovarian aging are limited. Hormone replacement therapy (HRT) and assisted reproductive technologies (ART) are the commonly employed treatments [53]. However, HRT not only magnifies the risk of hormone-related cancers, such as breast and ovarian cancer, but also heightens the risk of cardiovascular diseases [54, 55]. Moreover, due to the substantial reduction or depletion of ovarian reserve in aging ovaries, assisted reproductive techniques can only achieve limited effectiveness [56]. Our findings demonstrated that miR-143-3p directly bound to the 3' UTR of the UBE2E3 gene, leading to the inhibition of UBE2E3 expression and impacting the process of ovarian cell senescence. As such, interventions aimed at modulating miR-143-3p levels or activity could potentially alter the senescence-associated changes in ovarian cells and improve overall ovarian function. For example, pharmaceutical or gene therapy approaches may be designed to modulate miR-143-3p expression or activity in order to restore or enhance the expression of UBE2E3 and counteract ovarian senescence. These interventions could potentially rejuvenate ovarian tissues, improve oocyte quality, and increase fertility potential in women of advanced reproductive age. However, this requires more rigorous and detailed studies on the upstream regulatory mechanisms and specific molecular mechanisms of downstream targets of miR-143-3p. And further studies are needed to fully understand the broad implications of miR-143-3p regulation and its potential therapeutic applications for ovarian aging. For instance, the effects of interventions targeting miR-143-3p in the aged mice should be examined.”

Comment 4. To validate the direct targeting of UBE2E3 and LHCGR by miR-143-3p, the authors could perform luciferase reporter assays to confirm binding specificity and efficiency. Additionally, they could consider using CRISPR/Cas9 gene editing to generate UBE2E3 or LHCGR knockout GCs to confirm the role of these targets in GC senescence and hormone synthesis.

Response: Thank you for the reviewer's suggestions. In our study, to demonstrate direct regulation of LHCGR or UBE2E3 by miR-143-3p, we performed a dual-luciferase reporter assay. As shown in Figure 4B and Figure 5B, luciferase expression from the wild type but not from mutant constructs was significantly suppressed by miR-143-3p, indicating that miR-143-3p directly regulates LHCGR and UBE2E3 by binding to 3' UTR of LHCGR and UBE2E3. Subsequently, we used small RNA interference technology for transient knockdown of UBE2E3 and LHCGR mRNA and demonstrated the role for these targets in GC aging and hormone synthesis. 

As suggested by the reviewer, CRISPR/Cas9 gene editing is a useful tool for confirming the role of these targets in GC senescence and hormone synthesis. We will be very pleased to use the CRISPR/Cas9 gene editing technique in the further investigations. Once again, thank you very much for your kind comments and constructive suggestions.

Comment 5. To explore the potential therapeutic implications of their findings, the authors could test the effects of interventions targeting miR-143-3p in animal models of ovarian aging. For example, they could use miRNA inhibitors or other small molecules to reduce miR-143-3p expression and evaluate the effects on ovarian function and fertility.

Response: We appreciate the reviewer’s suggestion to explore the therapeutic implications of our findings by testing the effects of interventions targeting miR-143-3p in animal models of ovarian aging. Unfortunately, we are unable to conduct the miR-143-3p intervention experiments as you have suggested in the current project. This is primarily due to the long breeding cycle of aged mice, which poses challenges for timely completion of the project. We kindly ask for your understanding in this matter. Nevertheless, we highly value your suggestion and assure you that we will take it into consideration for future research. Exploring the potential therapeutic applications of miR-143-3p is important to us, and we will strive to incorporate it into our laboratory’s subsequent studies. If possible, we also aim to collaborate with other laboratories to better evaluate the effects of miR-143-3p intervention in animal models of ovarian aging. We sincerely thank you for your valuable suggestion and your interest in our research. If you have any further suggestions or comments, we would be delighted to hear them.

In the revised manuscript, we discussed this issue in the “Discussion section”. Please check line 393-396.

“And further studies are needed to fully understand the broad implications of miR-143-3p regulation and its potential therapeutic applications for ovarian aging. For instance, the effects of interventions targeting miR-143-3p in the aged mice should be examined.”

Comment 6. To better understand the molecular mechanisms involved in miR-143-3p regulation of GC senescence, the authors could perform RNA-sequencing (RNA-seq) and proteomic analyses to identify downstream targets and signaling pathways affected by miR-143-3p. Also, The authors can do the IPA analysis as well.

Response: Thank you for your valuable suggestions for improving the quality of our work. In this study, we primarily focused on the functional and preliminary mechanistic elucidation of miR-143-3p in GC senescence. As you pointed out, further studies are needed to understand the implications of our findings in the context of potential therapeutic targets for ovarian aging. Therefore, we completely agree with you on the importance of performing RNA-sequencing (RNA-seq) and proteomic analyses to identify downstream targets and signaling pathways affected by miR-143-3p in GC senescence. By incorporating these suggested experimental approaches, we believe that we will be able to provide a more comprehensive and mechanistic understanding of how miR-143-3p regulates GC senescence. We appreciate your insight and will make sure to include these analyses in our study.

Reviewer 2 Report

In this study, the authors investigated an association between age-related fertility decline in female mice and elevated expression levels of miR-143-3p. They concluded that miR-143-3p plays an essential role in senescence and steroid hormone synthesis in granulosa cells, suggesting its potential as a therapeutic target for ovarian aging interventions.

I read the study with interest. The study is well designed and well written and of importance. However, several issues should be addressed/corrected before a positive decision is made. My concerns are as follows:

1. Paragraph 4.1 - Please provide the age in weeks and the average body weight of the animals used in the study

2. Paragraph 4.1 - The authors also need to indicate the number and distribution of animals used in this experiment. They should also indicate how this sample was calculated.

3. The authors stated that all animal experiments were performed in accordance with the National Institute of Health guidelines for animal care and use and were approved by the Institutional Animal Care and Use Committee of Jinan University. Please include the approval number and date of issuance.

4. Supplementary tables and figures are important. Please include them as regular tables in the paper, as there is no restriction on tables and figures in MDPI journals.

5. Please indicate which statistical test was used to check the normality of the data distribution. If the data were not normally distributed, please indicate why the median with IQR was not used.

6. At the end of the discussion, the authors should indicate the limitations of this study and perspectives for future work.

Author Response

Reviewer #2: In this study, the authors investigated an association between age-related fertility decline in female mice and elevated expression levels of miR-143-3p. They concluded that miR-143-3p plays an essential role in senescence and steroid hormone synthesis in granulosa cells, suggesting its potential as a therapeutic target for ovarian aging interventions.

I read the study with interest. The study is well designed and well written and of importance. However, several issues should be addressed/corrected before a positive decision is made. My concerns are as follows:

Comment 1. Paragraph 4.1 – Please provide the age in weeks and the average body weight of the animals used in the study

Response: Thank you for pointing out this issue. We are very sorry for our negligence. We have added information about the age and weight of the mice in the “Animals” section (Revised manuscript, line 404-415).

“C57BL/6J female mice (3 to 15 months old) and KM female mice (3 weeks old) were purchased from the Experiment Animal Center of Guangdong Province, China, and maintained under 12 h light/dark cycle at controlled temperature (24°C ± 2°C) with relative humidity of 50% - 60%. Standard rodent diet and drinking water were freely accessible. Prior to the study, the C57BL/6J female mice were acclimatized for a minimum of one week. In the first part of our study, to investigate the relationship between age and miR-143-3p in mice, female mice were divided by age into one of the following groups (n = 6): young group (4 months old, 23 g average body weight), middle-aged group (8 months old, 28 g average body weight), and an old group (15-months old, 37 g average body weight). All animal experiments were conducted in accordance with the National Institute of Health Guidelines for the care and use of animals and approved by the Institutional Animal Care and Use Committee of Jinan University.”

Comment 2. Paragraph 4.1 – The authors also need to indicate the number and distribution of animals used in this experiment. They should also indicate how this sample was calculated.

Response: Thank you for your comment. We apologize for the lack of information regarding the number and distribution of animals used in our experiment. We have included this information in the revised manuscript to provide a more complete description in “Animal” section. Please check line 404-415.

Comment 3. The authors stated that all animal experiments were performed in accordance with the National Institute of Health guidelines for animal care and use and were approved by the Institutional Animal Care and Use Committee of Jinan University. Please include the approval number and date of issuance.

Response: Thank you for your suggestion. We have indicated the approval number and date of issuance in “Institutional Review Board Statement” section. (Revised manuscript, line 531-533).

Institutional Review Board Statement: The animal study protocol was approved by the Institutional Animal Care and Use Committee of Jinan (approval number: IACUC; issue No: 20210629-06, approval date: 29 June 2021).”

Comment 4. Supplementary tables and figures are important. Please include them as regular tables in the paper, as there is no restriction on tables and figures in MDPI journals.

Response: We appreciate you bringing this issue to our attention. As your suggestion, we have included the supplementary tables/figures as regular tables/figures and made necessary adjustments to the figures in the revised manuscript. Please check Table 1 and Table 2/Figure 1, Figure 2, and Figure 6.

Comment 5. Please indicate which statistical test was used to check the normality of the data distribution. If the data were not normally distributed, please indicate why the median with IQR was not used.

Response: We apologize for the omission and have revised the " Statistical Analysis " section accordingly in the revised manuscript. Please check line 505-510.

“The statistical analyses were performed with GraphPad Prism 8.0 soft-ware. All data were shown to fit a normal distribution by using Shapiro–Wilk test. The results are shown as the mean values ± 1 SD. Unpaired student’s t test was used to analyze the data between two groups. One-way ANOVA was used to determine significant intergroup differences. P-values less than 0.05 were considered significant and were marked as follows: P < 0.05 (*), P < 0.01 (**), P < 0.001 (***) and P < 0.0001 (****).”

Comment 6. At the end of the discussion, the authors should indicate the limitations of this study and perspectives for future work.

Response: We appreciate your valuable suggestions for improving the quality of our work. As your recommendation, we have added this content to the "Discussion" section. (Revised manuscript, line 391-396)

“However, this requires more rigorous and detailed studies on the upstream regulatory mechanisms and specific molecular mechanisms of downstream targets of miR-143-3p. And further studies are needed to fully understand the broad implications of miR-143-3p regulation and its potential therapeutic applications for ovarian aging. For instance, the effects of interventions targeting miR-143-3p in the aged mice should be examined”

Round 2

Reviewer 1 Report

The Authors made possible corrections to the current version of this manuscript which is now suitable to accept.